# Enantioselective total syntheses of (+)-stemofoline and three congeners based on a biogenetic hypothesis

Xiong-Zhi Huang[1], Long-Hui Gao [1] & Pei-Qiang Huang [1✉]

The powerful insecticidal and multi-drug-resistance-reversing activities displayed by the stemofoline group of alkaloids render them promising lead structures for further development as commercial agents in agriculture and medicine. However, concise, enantioselective total syntheses of stemofoline alkaloids remain a formidable challenge due to their structural complexity. We disclose herein the enantioselective total syntheses of four stemofoline alkaloids, including (+)-stemofoline, (+)-isostemofoline, (+)-stemoburkilline, and (+)-(11S,12R)-dihydrostemofoline, in just 19 steps. Our strategy relies on a biogenetic hypothesis, which postulates that stemoburkilline and dihydrostemofolines are biogenetic precursors of stemofoline and isostemofoline. Other highlights of our approach are the use of Horner–Wadsworth–Emmons reaction to connect the two segments of the molecule, an improved protocol allowing gram-scale access to the tetracyclic cage-type core, and a Cu-catalyzed direct and versatile nucleophilic alkylation reaction on an anti-Bredt iminium ion. The synthetic techniques that we developed could also be extended to the preparation of other *Stemona* alkaloids.

[1] Department of Chemistry, Fujian Provincial Key Laboratory of Chemical Biology, College of Chemistry and Chemical Engineering, Xiamen University, Xiamen, Fujian 361005, P.R. China. ✉email: pqhuang@xmu.edu.cn

Nature continues to serve as an invaluable source for the development of pharmaceutical drugs[1], agrochemical agents[2], and biomedical probes[3]. In this context, efficient total synthesis of these bioactive natural products is of crucial importance[4]. With significant advances in synthetic methodologies and strategies, many artful instances of total synthesis have been documented in the last few decades[4]. However, most of these reported studies involved only one target molecule[5]. Considering the huge number of biologically interesting natural products, there is clearly unmet need in developing novel synthetic strategies that can branch out toward multiple target compounds. In recent years, multi-target-oriented synthetic approaches, such as unified strategy[6–9], collective synthesis[10,11], and diversity-oriented synthesis[12], have emerged as a frontier. It should be noted that many of these methods are biomimetic or bioinspired in that they rely on known biogenetic routes or hypotheses. But for those natural products whose biogenetic pathways are unknown or have not been investigated in detail, it is necessary to develop biogenetic hypotheses[13,14].

The *Stemona* alkaloids are a class of structurally diverse natural products that act as the principle active constituents in *Stemona* plants (Stemonaceae)[15–20]. Known as "Bai Bu" in traditional Chinese medicine, *Stemona* plants have been used in East Asia for thousands of years as insecticides and anti-cough agents[15,16,19,20]. To date, more than 215 *Stemona* alkaloids have been isolated[15], classified by Pilli[19,20] into eight structural groups (see Supplementary Fig. 1). The attractive bioactivities[15,16] and unique polycyclic structures of these alkaloids have led to intense phytochemical, synthetic, and biomedical investigations, and many innovative total syntheses have been achieved[16,18–24].

The stemofoline group of alkaloids (Fig. 1) are a highly versatile set of compounds that represent promising lead structures for agricultural and medicinal applications[15,16]. Stemofoline (1), the parent member of the stemofoline group, was isolated first from *Stemona japonica* by Irie and coworkers in 1970[25], and then from several other *Stemona* species. Its structure, including the absolute configuration, was determined by single-crystal X-ray crystallographic analysis[25]. To date, more than twenty members of the stemofoline group have been isolated and structurally determined, including isostemofoline (2)[26], didehydrostemofoline (3)[27,28], isodidehydrostemofoline (4)[27], methoxystemofoline (5, suggested structure)[29],

methylstemofoline (6)[30], stemoburkilline (7)[31,32], and (11S,12R)-dihydrostemofoline (8)[31] (Fig. 1). The high insecticidal activity[33] of stemofoline (1) was first reported in 1978, according to which the compound acts as a potent agonist of the insect nicotinic acetylcholine receptors (nAChR) (EC$_{50}$ = 1.7 nM) and is associated with acetylcholinesterase (AChE) inhibition[15]. Ye and coworkers have established that the cage-type moiety of stemofoline is pivotal to its insecticidal activity[26]. Notably, by using stemofoline (1) as a lead, synthetic cyanotropanes have been developed as a class of commercial insecticides[2]. Moreover, stemofoline (1) has been found to alleviate inflammation[34] and reverse multidrug resistance of certain types of cancer[15,35–37]. Indeed, it has been shown that stemofoline (1) increases the sensitivity of patients toward anticancer drugs such as vinblastine, paclitaxel, and doxorubicin[15,35]. Once again, this desirable properties can be attributed to the core cage structure with nonpolar side chains[35].

Much efforts have been devoted to the synthesis of stemofoline alkaloids. Starting from stemofoline[38] and (11Z)-1′,2′-didehydrostemofoline[39,40], several semisynthetic stemofoline alkaloids and analogues have been prepared and screened by Ye et al. and Pyne et al., respectively. Although tremendous efforts have been devoted to the total synthesis of these alkaloids[41–47], successful examples remain rare, including racemic total syntheses of isostemofoline (2) by Kende[48] and didehydrostemofoline (asparagamine A, 3) and isodidehydrostemofoline (4) by Overman[49], and enantioselective formal total syntheses of didehydrostemofoline and isodidehydrostemofoline by Martin[50]. Notably, although stemofoline (1) displays remarkable bioactivities, and has proven to be a promising lead for the development of insecticide and anticancer agents, its total synthesis has not yet been achieved.

As a part of our efforts to achieve efficient total synthesis of structurally complex, enantiopure alkaloids[51], we have been focused in the last decade on developing versatile methodologies for the direct transformation of amides[52,53]. We have recently reported a method for the construction of tropinone ring systems[54] and applied it to the total synthesis of (+)-methoxystemofoline (5)[55]. We disclose herein biogenetic hypothesis-based syntheses of (+)-stemofoline (1), (+)-isostemofoline (2),

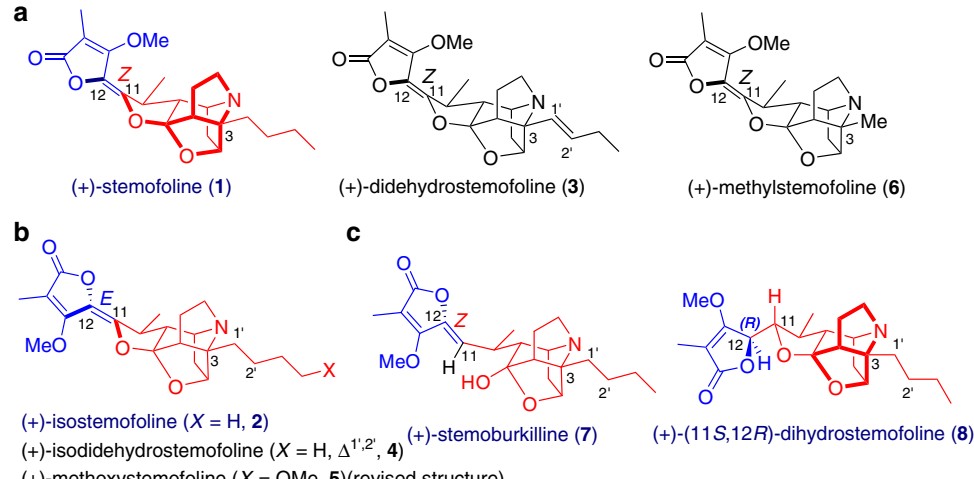

**Fig. 1 Structures of representative stemofoline alkaloids. a** Z-Series: (+)-stemofoline (1); (+)-didehydrostemofoline (3); (+)-methylstemofoline (6). **b** E-Series: (+)-isostemofoline (2); (+)-isodidehydrostemofoline (4); (+)-methoxystemofoline (5) (revised structure). **c** 11,12-Dihydro series: (+)-stemoburkilline (7); (+)-(11S, 12R)-dihydrostemofoline (8).

**Fig. 2 Our biogenetic hypothesis and retrosynthetic analysis of stemofoline alkaloids. a** Our biogenetic hypothesis. **b** Retrosynthetic analysis based on our biogenetic hypothesis.

(+)-stemoburkilline (**7**), (11S,12R)-dihydrostemofoline (**8**), and diastereomer **9**.

## Results

**Biogenetic hypothesis**. Structurally, stemofoline alkaloids consisted of a tetronate moiety connected through an ene diether (C11–C12) to a cage-like aza-pentacyclic ring system. Both the enantioselective construction of the cage-like aza-pentacyclic ring system[56] and the stereoselective formation of the ene diether moiety are challenging[48,49,55]. Indeed, our previous total synthesis was lengthy and contained several steps of reaction with low chemo-, regio-, and/or diastereoselectivity[55]. In addition, the stemofoline alkaloids display great variation in the oxidation state and connectivity at C11, the stereochemistry of the ene diether connector, and substitution at the bridgehead carbon C3. Taking these factors into consideration, we wanted to develop a concise, versatile synthetic strategy. Although a bioinspired approach is highly desirable[57], and indeed there are already several hypotheses on the origin of the central building

blocks in *Stemona* alkaloids[15–17,28], the basic biogenetic steps in the formation of *Stemona* alkaloids remain unknown[15]. In the search for a biogenetic relationship between the stemofoline group of alkaloids, we have noted that isostemofoline (**2**)[26], stemoburkilline (**7**), and (11S,12R)-dihydrostemofoline (**8**)[31,32] were congeners of stemofoline (**1**), and Pyne et al. also observed that (11S,12S)-dihydrostemofoline (**9**) (Fig. 2a) could be converted into a mixture of stemoburkilline (**7**) and (11S,12R)-dihydrostemofoline (**8**) upon treatment with DBU[32]. Moreover, Ye and coworkers[26] have reported the co-existence of stemofoline (**1**) and isostemofoline (**2**), the latter as a minor congener, in *S. japonica*, whereas Jiwajinda *et al*. has detected equal amounts of 11Z-didehydrostemofoline (**3**) and 11E-didehydrostemofoline (**4**) in *S. collinsae*[27]. Combining these findings with ours, we hypothesized that the known compounds stemoburkilline (**7**) and (11S,12R)-dihydrostemofoline (**8**), and possibly the unknown compound (11S,12S)-dihydrostemofoline (**9**), are plausible biogenetic precursors of stemofoline (**1**) and isostemofoline (**2**) (Fig. 2a).

**Fig. 3 Enantioselective synthesis of the tropinone building block 16.** Reagents and conditions: **a** 19, MeOH, rt, 3 d, 90%; **b** (COCl)$_2$, DMSO, Et$_3$N, CH$_2$Cl$_2$, −78 °C; **c** NaH, 21, THF, 0 °C to reflux, 60% (for 2 steps); **d** TMSOTf, Et$_3$N, CH$_2$Cl$_2$, 0 °C, **e** Tf$_2$O, DTBMP, ZnBr$_2$, CH$_2$Cl$_2$, −78 °C to rt, 78% (for 2 steps from cis-**17**).

**Retrosynthetic analysis**. Guided by our biogenetic hypothesis (Fig. 2a), we conducted a retrosynthetic analysis of stemofoline (**1**) as depicted in Fig. 2b. The (Z)-5-(dihydrofuran-2(3H)-ylidene)-4-methoxy-3-methylfuran-2(5H)-one [(Z)-ene diether moiety could be constructed through C–H oxygenation of stemoburkilline (**7**). The functionalized C11–C12 alkene moiety in **7** could be assembled through Horner–Wadsworth–Emmons-type reaction of diethyl (3-methoxy-4-methyl-5-oxo-2,5-dihydrofuran-2-yl)phosphonate (**10**) with the pentacyclic cage-type hemiacetal **11**. Other key issues to be addressed were: (1) the O-debenzylation of **13** triggering double cyclization, (2) the direct, chemo-, regio-, and diastereoselective methylation at C10 to build a chiral canter, and (3) direct installation at C3 a butyl group or other alkyl groups needed for the synthesis of other stemofoline alkaloids and analogues.

**Syntheses of stemoburkilline (7), its congeners 8, and 9**. To implement our strategy, we first developed an improved five-step protocol to furnish the tropinone building block **16** on a multi-gram scale (Fig. 3). To this end, commercially available α-benzyloxy-γ-lactone (S)-**18** was treated with O-silylated β-aminoethanol **19** in methanol at room temperature for 2–3 days, which afforded γ-hydroxyamide **20** in 90% yield. Swern oxidation of **20** generated a diastereomeric mixture of hemiaminals, which was then subjected without purification to NaH-mediated Horner–Wadsworth–Emmons reaction with dimethyl (2-oxopropyl)phosphonate (**21**) followed by cyclization to produce the desired cis-ketolactam **17** in 60% yield (over two steps), along with the trans-diastereomer in 21% yield. Conversion of **17** to bromo-tropinone building block **16** was achieved in 78% yield via a two-step keto-lactam cyclization-bromination cascade (TMSOTf, Et$_3$N, CH$_2$Cl$_2$, 0 °C; Tf$_2$O, DTBMP, ZnBr$_2$, CH$_2$Cl$_2$, −78 °C to rt) that we previously developed[54].

We next focused on the development of a direct and versatile method for the installation of an n-butyl group on the bridgehead carbon of **16**. In our previous strategy[55], three steps were required to introduce the 4-methoxybutyl group. A survey of literatures indicated that the bridgehead nitrogen in similar but simpler ring systems, such as 1-chloro-9-methyl-9-azabicyclo[3.3.l]nonane, enhances the rate of solvolysis dramatically[58]. However, attempted direct alkylation of a similar α-chloroamine with organolithium or Grignard reagent produced disappointing results[59,60]. Encouragingly, Kibayashi and coworkers have reported that bridged tricyclic N,O-acetals can be alkylated using Grignard reagents in the presence of Et$_2$AlCl through S$_N$1 reaction on bridgehead anti-Bredt iminium ion

intermediates[60,61]. In light of these precedents, we opted to explore whether metal-catalyzed direct alkylation of 3-bromotropinone building block **16** with Grignard reagents was feasible. After extensive screening (cf. Table 1), we achieved gram-scale coupling of **16** with n-butyl magnesium bromide in THF to afford the desired butylated product **15** in 81% yield, with CuCl$_2$ as a catalyst, TMEDA (N,N,N′,N′-tetramethylethylenediamine) as ligand and LiOMe as additive (Table 1, entry 12). According to the precedent set by Kibayashi, it is possible that the alkylation reaction involves an S$_N$1 mechanism via the intermediacy of anti-Bredt bridgehead iminium ion **A**. Significantly, the method can be extended to other Grignard reagents, thereby allowing the introduction of other desired simple or functionalized alkyl groups at C3. In this manner, alkylated products **15a**–**15d** (Fig. 4) could be prepared directly from **16** in good yields (75–85%).

To construct compound **14**, which contains a tricyclic core with an ester-enone moiety (Fig. 5), tropinone **15** was successively treated with LDA and ethyl glyoxalate (**22**) to yield aldol adducts **23-1** and **23-2** in 71% yield with a diastereomeric ratio of 1.1:1, along with regioisomer **23a** in 15% yield. The lack of diastereoselectivity was of no consequence as dehydration of **23-1** and **23-2** afforded the desired enone **24** as a single isomer in 72% yield. NOESY studies indicated that **24** adopted an E configuration (see Supplementary Fig. 3). From these experimental results, we deduced that the structures of two diastereomers **23-1** and **23-2** have the stereochemistries shown in Fig. 5.

Desilylation of **24** with p-TsOH, followed by bromination of the resultant alcohol **25**, furnished tropinone halide **26** in an overall yield of 91.2% (for 2 steps; Fig. 5). Exposure of **26** to EtONa in THF resulted in the formation of tricyclic enone **14** in excellent yield (95%). Another key step for our strategy resided in the regio- and diastereoselective methylation at C10. To our delight, the intended conjugate addition was achieved by treating **14** with MeLi-DMPU in diethyl ether[62], leading to **13** as a single regio- and diastereomer in 86% yield. Pd/C catalyzed O-debenzylation of **13** in ethanol was accompanied by tandem lactolization, which furnished the cage-like core structure **27** directly as a single diastereomer in 87% yield. Exposing **27** to p-TsOH in toluene produced the key pentacyclic lactone **12** in 93% yield. Importantly, neither the hydrogenation nor lactonization step required purification to yield **12** of sufficient purity for further reactions. The stereochemistry of **12** was determined by NOESY experiments (see Supplementary Fig. 6), which confirmed that the addition of MeLi to **14** was instrumental in creating the two vicinal chiral centers at C9 and C10.

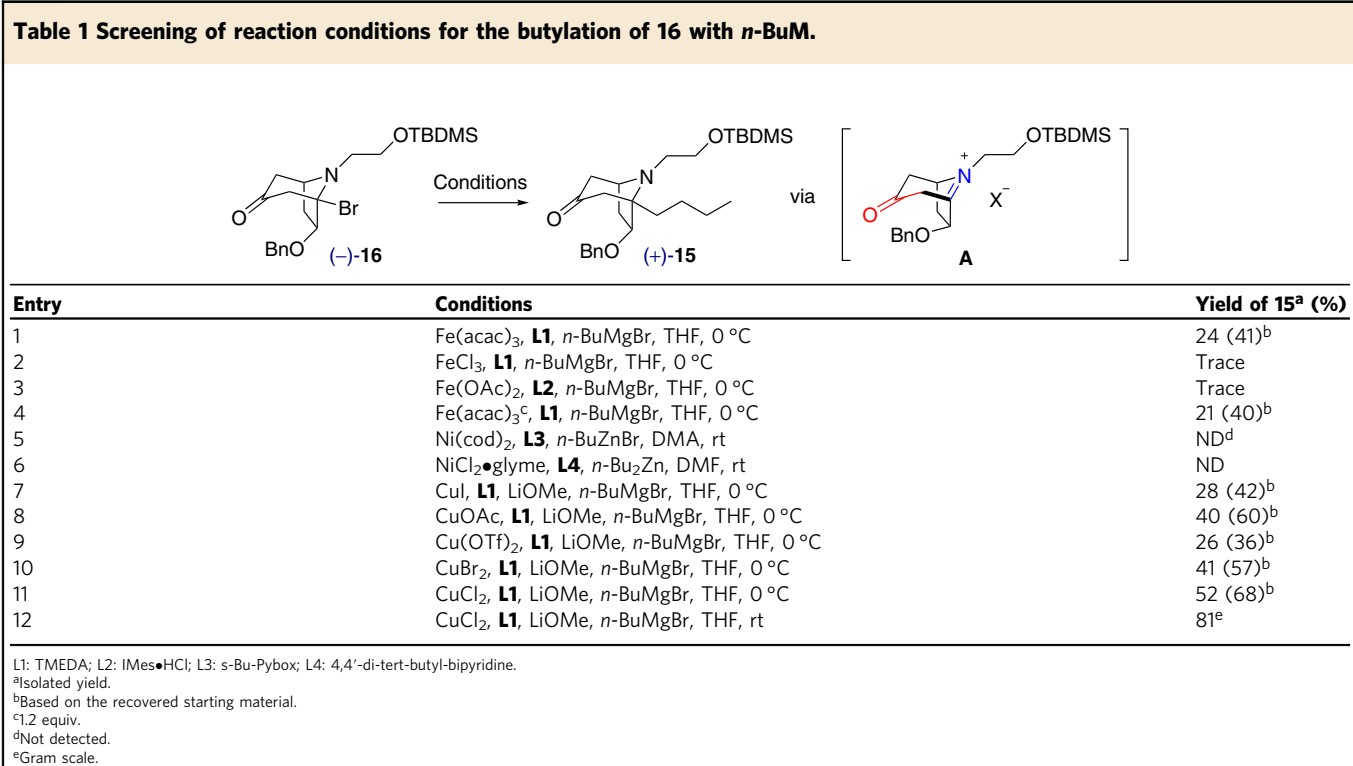

**Table 1 Screening of reaction conditions for the butylation of 16 with *n*-BuM.**

| Entry | Conditions | Yield of 15[a] (%) |
|---|---|---|
| 1 | Fe(acac)₃, **L1**, *n*-BuMgBr, THF, 0 °C | 24 (41)[b] |
| 2 | FeCl₃, **L1**, *n*-BuMgBr, THF, 0 °C | Trace |
| 3 | Fe(OAc)₂, **L2**, *n*-BuMgBr, THF, 0 °C | Trace |
| 4 | Fe(acac)₃[c], **L1**, *n*-BuMgBr, THF, 0 °C | 21 (40)[b] |
| 5 | Ni(cod)₂, **L3**, *n*-BuZnBr, DMA, rt | ND[d] |
| 6 | NiCl₂•glyme, **L4**, *n*-Bu₂Zn, DMF, rt | ND |
| 7 | CuI, **L1**, LiOMe, *n*-BuMgBr, THF, 0 °C | 28 (42)[b] |
| 8 | CuOAc, **L1**, LiOMe, *n*-BuMgBr, THF, 0 °C | 40 (60)[b] |
| 9 | Cu(OTf)₂, **L1**, LiOMe, *n*-BuMgBr, THF, 0 °C | 26 (36)[b] |
| 10 | CuBr₂, **L1**, LiOMe, *n*-BuMgBr, THF, 0 °C | 41 (57)[b] |
| 11 | CuCl₂, **L1**, LiOMe, *n*-BuMgBr, THF, 0 °C | 52 (68)[b] |
| 12 | CuCl₂, **L1**, LiOMe, *n*-BuMgBr, THF, rt | 81[e] |

L1: TMEDA; L2: IMes•HCl; L3: s-Bu-Pybox; L4: 4,4′-di-tert-butyl-bipyridine.
[a]Isolated yield.
[b]Based on the recovered starting material.
[c]1.2 equiv.
[d]Not detected.
[e]Gram scale.

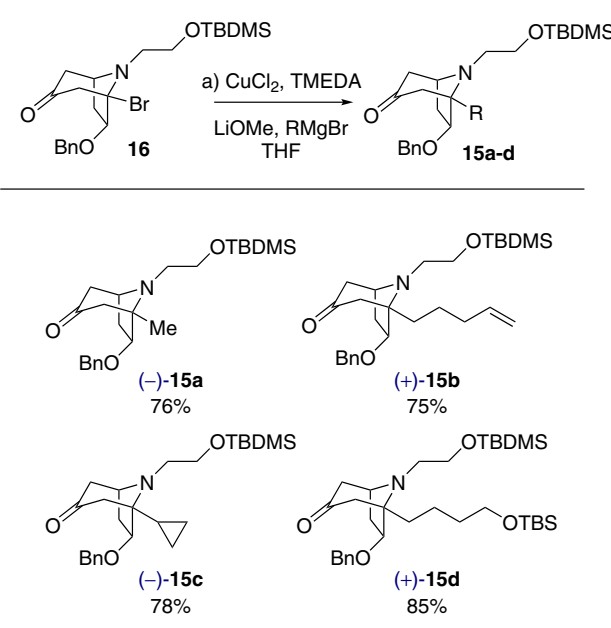

**Fig. 4 Direct alkylation of 16 with Grignard reagents.** Reagents and conditions: **a** CuCl₂, TMEDA, LiOMe, RMgBr, THF, rt, 30 min.

After **12** was synthesized, it was reduced with DIBAL-H to afford hemiacetal **11** as an inseparable diastereomeric mixture in 78% yield (Fig. 6). Once again, the lack of diastereoselectivity did not matter because treatment of **11** with potassium *t*-butoxide and diethyl (3-methoxy-4-methyl-5-oxo-2,5-dihydrofuran-2-yl) phosphonate (**10**), prepared in one-pot from γ-bromotetronate[63], resulted in the formation of stemoburkilline (**7**), (11*S*,12*R*)-dihydrostemofoline (**8**), and the diastereomeric (11*S*,12*S*)-dihydrostemofoline (**9**), in 27%, 24%, and 33% yield, respectively (combined yield: 84%). ¹H and ¹³C nuclear magnetic resonance

(NMR) and polarimetric data of **7**–**9** were largely consistent with previously reported values (see Supplementary Tables 5–10)[31,32].

Given that the treatment of (11*S*,12*S*)-dihydrostemofoline (**9**) with DBU yielded a tautomeric mixture of **9**, (11*S*,12*R*)-dihydrostemofoline (**8**), and stemoburkilline (**7**) in a ratio of 39:37:24[32], we speculated that stemoburkilline (**7**) could be a kinetic tautomer and that, under kinetic conditions, it would be possible to selectively convert both **8** and **9** into **7**. Indeed, treatment of a mixture of **8** and **9** with LHMDS resulted in the formation of stemoburkilline (**7**) in 98% yield (Fig. 6).

**Total syntheses of stemofoline (1) and isostemofoline (2).** To complete the total synthesis of stemofoline, a one-step Wacker-type reaction of stemoburkilline (**7**) was explored. Unfortunately, none of the reaction conditions (Na₂PdCl₄, TBHP[64,65]; PdCl₂, CuCl, O₂[66,67]; Pd(TFA)₂, IMes[68]) (see Supplementary Table 1) that we tested yielded the desired final product stemofoline (**1**). In each instance, a complex mixture of by-products, decomposition of the starting material, or lack of any major reaction whatsoever, was observed.

Next, we envisioned a two-step dehydrogenation of **8** and **9** consisting of introduction of a leaving group at C12 followed elimination. However, oxygenation with IBX/DMSO or DBU/O₂, halogenation with NIS/NBS, or methylsulfidation with MeSSMe, are all found to be disappointingly inadequate (see Supplementary Table 2). Attempted conversion of **8** and **9** to **1** (see Supplementary Table 3) through cyclization to form **29** followed by oxidation was also unsuccessful because treating a mixture of **8** and **9** with LHMDS and TMSCl resulted in no cyclization and exclusive formation of an *O*-TMS derivative **30** in 93% yield (Fig. 7a). We then turned our attention to the electrophilic cyclization of **7**. However, **7** apparently showed no reaction with NIS and was thus mostly recovered (>95%), regardless of whether CH₂Cl₂, THF or MeCN was used as solvent, and whether the cyclization was performed at room temperature or under reflux. Running the reaction in the presence of a base (NEt₃, DBU,

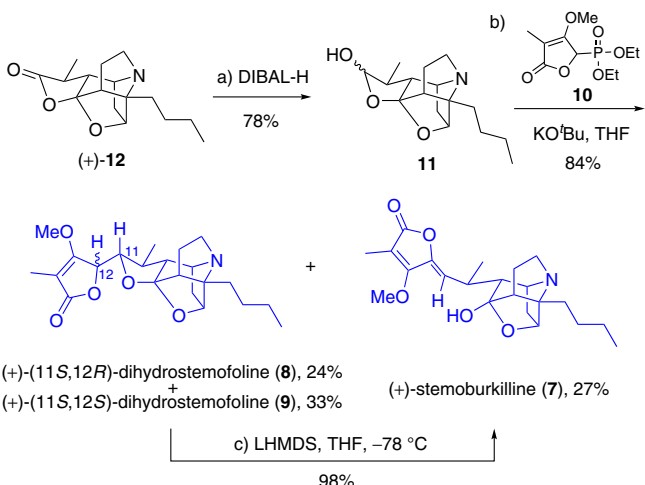

**Fig. 5 Synthesis of the pentacyclic lactone 12.** Reagents and conditions: **a** LDA, ethyl glyoxalate (**22**), −78 °C, 1.5 h, 71% for **23** and 15% for **23a**; (**b**) CDI, *i*-Pr₂NEt, DMAP, CH₂Cl₂, 0 °C to rt, 4 h, 72%; **c** *p*-TsOH, acetone, 50 °C, 30 min, 96%; **d** CBr₄, PPh₃, CH₂Cl₂, 0 °C, 30 min, 95%; **e** EtONa, THF, 0 °C, 20 min, 95%; **f** MeLi, DMPU, Et₂O, −40 °C, 20 min, 86%; **g** Pd/C, H₂, EtOH, rt, 2 d, 87%; **h** *p*-TsOH, toluene, 85 °C, 2 h, 93% (83% for two steps from **13** without further purification).

**Fig. 6 Completion of the total syntheses of stemoburkilline (7), (11S,12R)-dihydrostemofoline (8), and (11S,12S)-dihydrostemofoline (9).** Reagents and conditions: **a** DIBAL-H, CH₂Cl₂, −78 °C, 30 min, 78%; **b** *t*-BuOK, **10**, THF, 0 °C, 30 min, 84% (27% for **7**; 24% for **8**; 33% for **9**); **c** LHDMS, THF, −78 °C, 1 h, 98%.

NaOEt, or NaH) led to the formation of a mixture of **7**–**9** (see Supplementary Table 4). Interestingly, exposing **7** to I₂/NaHCO₃ in THF at room temperature produced the unexpected iodoetherification product **31** as a single diastereomer in 86% yield (Fig. 7b). A plausible mechanism for the diastereoselective formation of **31** is depicted in Fig. 7b[69].

Finally, a dibromination tactic[70] was explored (Fig. 8). To avoid possible interference of the hydroxyl group, **7**–**9** were converted to TMS-protected **30** in 95% yield. Compound **30** reacted with Br₂ regioselectively at C11–C12 alkene to afford **32** as a diastereomeric mixture. Unexpectedly, subsequent treatment of this mixture with DBU in CH₂Cl₂ caused it to revert back to **30**. Finally, we were pleased to find that treating **32** with TBAF in THF yielded stemofoline (**1**) directly in 50% yield, isostemofoline (**2**) in 10% yield, and stemoburkilline (**7**) in 12% yield. ¹H and ¹³C NMR and polarimetric data of stemofoline (**1**) {[α]ᴅ²⁵ +276 (*c* 1.0, MeOH); lit.²⁵ [α]ᴅ²⁵+273 (MeOH); lit.²⁸ [α]ᴅ²⁵ +270 (*c* 0.8, MeOH)}, and isostemofoline (**2**) {[α]ᴅ²⁵ +102.1 (*c* 0.5, CHCl₃)} were fully consistent with those reported in earlier studies (see Supplementary Tables 11–13), which confirmed not only the structures including relative and absolute stereochemistries of our synthetic stemofoline (**1**) and isostemofoline (**2**), but also those of stemoburkilline (**7**), (11S,12R)-dihydrostemofoline (**8**), and (11S,12S)-dihydrostemofoline (**9**).

## Discussion

In summary, we have achieved the total synthesis of stemofoline (**1**) after it was first isolated and structurally characterized 50 years ago. Our method features a relatively concise (19 steps) route that leads to multiple biomedically relevant compounds in an enantioselective manner. Particularly, we were able to simultaneously obtain stemofoline (**1**) and isostemofoline (**2**), two other members belonging to the same group stemoburkilline (**7**), (11S,12R)-dihydrostemofoline (**8**), as well as the (11S,12S)-diastereomer **9** at step 16 of the total synthesis. We argue that our success could be, at least in part, contributed to, and confirm, our biogenetic hypothesis. Furthermore, the synthetic efficiency that we accomplished was also

**Fig. 7 Attempted transformations of 7–9 into 1. a** Reaction of **8** and **9** with TMSCl. **b** Reaction of **7** with I$_2$.

**Fig. 8 Completion of the total syntheses of stemofoline (1) and isostemofoline (2).** Reagents and conditions: **a** LHMDS, TMSCl, THF, −78 °C, 1 h, 95%; **b** Br$_2$, CH$_2$Cl$_2$, rt, 30 min; **c** TBAF, THF, 0 °C, 1 h, 50% for **1**; 10% for **2**; 12% for **7** (for 2 steps from **30**).

ensured by: (1) an improved three-step protocol that allowed the gram-scale synthesis of keto-lactam *cis*-**17**, (2) a Cu-catalyzed direct nucleophilic alkylation reaction of 3-bromotropinone **16**, (3) a concise eight-step route for the efficient and selective construction of the pentacyclic core **12**, (4) the successful use of Horner–Wadsworth–Emmons reaction to assemble the two segments of the target molecules, and (5) the convergent transformation of **7–9** via **30** to biomimetically forge stemofoline (**1**) and isostemofoline (**2**). The realization of this bioinspired approach allowed us to predict that (11*S*,12*S*)-dihydrostemofoline (**9**) might be a natural product yet to be discovered. The direct, flexible, and versatile introduction of different side chains at C3 of **16** to form **15a–15d** opened an avenue for the synthesis of other stemofoline alkaloids such as methylstemofoline (**6**) and its analogues. Moreover, this strategy may also find application in the synthesis of other *Stemona* alkaloids.

## Methods

**General**. All reactions were performed anhydrously under nitrogen atmosphere. All reagents were purchased from commercial suppliers without further purification. Solvent purification was conducted according to Purification of Laboratory Chemicals (Peerrin, D.D.; Armarego, W.L. and Perrins, D.R., Pergamon Press: Oxford, 1980). Yields were calculated based on the weights of chromatographically isolated products. Reactions were monitored by thin-layer chromatography (TLC) on plates (GF254) supplied by Yantai Chemicals (China). The TLC spots were visualized under ultraviolet light or by staining with an ethanolic solution of phosphomolybdic acid and cerium sulfate or iodine vapor. Flash column

chromatography was performed using silica gel (200–300 mesh) from Qingdao Haiyang Chemicals. NMR spectra were recorded on Bruker AV III 400, Bruker AV III 500, Bruker AV III 850 instruments, and calibrated with tetramethylsilane (TMS) ($\delta$H = 0.00 ppm) and CDCl$_3$ ($\delta$C = 77.00 ppm) as internal references. Multiplicities were designated as follows: s = singlet, d = doublet, t = triplet, q = quartet, br = broad, dd = double doublet, td = triple doublet, dt = double triplet, dq = double quartet, m = multiplet. Infrared spectra were measured on a Nicolet FT-380 spectrometer using film KBr pellet techniques. High-resolution mass spectra analyses were performed on a Fourier transform ion cyclotron resonance mass spectrometer (Bruker Daltonics) with a 7-T magnet (Magnex) and an electrospray ionization source (Apollo II, Bruker Daltonics) under positive-ion mode. Optical rotations were measured on an Anton Paar MCP-500 polarimeter.

**Experimental data**. For detailed experimental procedures, see Supplementary Methods. NMR spectra were presented as Supplementary Figs. 11–66. For NMR comparison between the reference compounds and compounds that we synthetized in this study, including stemofoline (**1**), isostemofoline (**2**), stemoburkilline (**7**) and (11*S*,12*R*)-dihydrostemofoline (**8**), see Supplementary Tables 5–13. For ORTEP diagrams for compounds **30**, see Supplementary Fig. 13.

## Data availability

The X-ray crystallographic coordinates for structures reported in this article have been deposited at the Cambridge Crystallographic Data Centre (CCDC), under deposition numbers CCDC 1962403 for compound **30**. These data can be obtained free of charge from the Cambridge Crystallographic Data Centre via http://www.ccdc.cam.ac.uk/data_request/cif.

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

## Acknowledgements

We are grateful to the National Natural Science Foundation of China (22071204 and 21931010), the National Key R&D Program of China (2017YFA0207302), and the Program for Changjiang Scholars and Innovative Research Team in University of the Ministry of Education, China, for financial support. We thank Dr. J.L. Ye for assistance in the preparation of XRD documents and in checking the NMR data.

## Author contributions

P.-Q.H. conceived and directed the project and wrote the paper with assistance from X.-Z.H.; X.-Z.H. performed the experiments and analyzed the data; L.-H.G. improved the synthesis of **17**. All authors discussed the results and commented on the paper.

## Competing interests

The authors declare no competing interests.
