## [Peer Review File · Nature Communications]

REVIEWER COMMENTS

Reviewer #1 (Remarks to the Author):

In this manuscript, Huang and co-workers described their total syntheses of stemofoline and four related analogs including its plausible biogenetic precursors. This work is important and recommended for publication for the following reasons.

- i) Stemofoline and its analogs are important natural products with insecticidal and anticancer activity and they are very challenging target molecules for total synthesis.
- ii) Very few total syntheses of the closely related natural products were achieved. These prior total syntheses are elegant and remarkable, but there are also drawbacks in these syntheses. For example, the formation of the di-oxygenated tetrasubstituted double bond has been a long-standing challenge.
- iii) The synthetic approach described by Huang and co-workers are very different from the prior arts, therefore justifies for novelty.
- iv) They have developed a good solution to construct the problematic di-oxygenated tetrasubstituted double bond. Their protocol involves an HWE reaction followed by a bromine oxidation and a TBAF-promoted elimination and substitution. The HWE reaction allowed them to access two dihydrostemofoline analogs (8 and 9) and stemoburkilline (7), which could be further converted to stemofoline and isostemofoline. This result also provides evidence to support the proposed biogenetic hypothesis. This protocol will likely be adapted to make many other natural products with such a tetrasubstituted double bond in the stemona alkaloid family.
- v) A remarkable bridgehead bromide substitution via a sequence of elimination and addition is worth noting as well. This substitution allows the introduction of various alkyl group at the bridgehead carbon, which could be used to make other stemofoline analogs with varied alkyl group at this position.
- vi) Additionally, the authors identified a nice conjugate addition to solve the C10 stereochemistry problem, which troubled the previous syntheses.

With the above strength said, there are also a few minor weaknesses.

- i) Since the dihydrostemofoline analogs 8 and 9 and stemoburkilline (7) are inevitable intermediates on the way to stemofoline and isostemofoline is a minor product from the last step, the word "divergent" in the manuscript is not very accurate.
- ii) Is the stereochemistry of 23-1 and 23-2 different at C9 or C10? Why both can be converted to 24? An explanation is needed here.
- iii) Did the authors try any aerobic oxidation of 7, 8, or 9 to produce stemofoline?
- iv) The writing of this manuscript needs to be polished.

Reviewer #2 (Remarks to the Author):

The ms describes the asymmetric syntheses of four stemofoline-type alkaloids and a related diastereomer. The ms describes a substantial body of work aimed at synthesizing the tetracyclic core unit of the stemafoline ring system and then developing methodology to construct and install the butenolide onto the tetracyclic core. The linkage between the tetracyclic core and the butenolide moiety incorporates an ene diether motif.

Analytical and spectral data are consistent with structures of compounds.

Comments, questions, and clarifications which the authors might like to address are provided below.

Pg1, Abstract; Ln 11-15: Sentence starts off in plural form and then changes to singular form.

Ln 17: "biogenic" or biogenetic hypothesis?

Ln 21; "...to tackle the frustrated ene diether connector"..... meaning of frustrated in this context??

Ln 26; ..."Stemona alkaloids having similar structural features." Useful to specify what the structural features are.....tetracyclic cage-type structure?

Pg 2, Ln 52-55: "The attractive.....achieved." This section says almost the same thing as on pg 4, Ln 83-84: "The intriguing....synthetic targets.", thus somewhat repetitive.

Pg 4, Fig. 1, structure 8: stereocentres at C-11 and C-12 can be better drawn. Since C-11 is part of a envelope conformation, the H and butenolyl unit is better represented as pseudoaxial and pseudoequatorial, respectively. The bond to the methyl group should be a normal line. Throughout the ms (and SI), bold bonds and normal bonds are used interchangeably to indicate stereochemistry.

Pg 5, Ln 120-123: "...isostemofoline (2), stemoburkilline (7),.....DBU." This section is somewhat confusing as the authors talked about (11S,12R)-8 and mentioned Pyne's work involving (11S,12S)-9 (Fig 2a). Further down, Ln 129-130, the same compounds were mentioned again. In Fig. 2A (pg 7), compound 9 is shown with (11R,12R) configurations. Further, the (11S,12S)-9 in Fig 5 added to the confusion.

Pg 7, Fig. 2A: The schematic showing the connectivity between compounds 7, 8, and 9 and alkaloids 1 and 2 is "oxidative cyclization". Oxidative cyclization is applicable to 7  1 and 2; however, for 8/9 -- 1 and 2, the connection is "oxidation" (dehydrogenation).

Pg 8, Fig. 3: In the synthesis of tropinone 16, the % ratio of cis-17 : trans-17 (60:21) was lower than the previous reported route (72:20; ref 55). Did the authors recycle trans-17 to obtain more cis-17 or was the trans-diastereomer not used?

Pg 9, Table 1(a) and (b): The way the data were presented caused some confusion at first, as the Table footnote b was easily mistaken by (b) (part of Table 1; Ln 201) depicting "Direct....other (functionalized) Grignard reagents" The presentation of structures 15a-d is somewhat awkward and the authors might want to combine their data into one Table.

The chemistry, modeled along the studies of Kibayashi et al., is very nice. Kibayashi's azabicyclic N,O-acetal substrates are of slightly larger ring sizes and the generation of anti-Bredt iminium ions are not as geometrically constrained. The authors' compound 16 is more structurally rigid and formation of the anti-Bredt iminium ion A (Table 1a) most likely is higher energy demanding. Do the authors think that the reaction might proceed via an organocopper-type intermediate - e.g., Tropinone-16—Cu(L)—Grignard instead of iminium ion A?

Pg 10-11 and Fig. 4: Line 206-207; The aldol reaction of tropinone 15 and glyoxalate 22 leads to aldol product 23-1 and 23-2 and a regioisomer 23a (not shown) The structure numbering used here is quite confusing as subsequent reactions indicating compound 23 to mean 23-1 and 23-2 (line 209). Also, 23a is labeled 23b in SI (S68/69). Does the 1.1:1 diastereomeric mixture indicate that the product is epimeric only at the carbinol center, and the α -carbon stereocentre is confirmed on account of the axial α -C-C(OH) bond?

Line 214: "...overall yield of 91.2%.." In Fig. 4, the yield shown is 95%.

Line 223: "The two transformations....83% yield." Not exactly one step. The experimental procedure in SI suggests that the two steps were executed sequentially and involved a work-up step to obtain crude 27 which was used without purification in the lactonization step.

Line 225-226: "...which confirmed that....vicinal chiral centers at C9 and C10." In the conversion of 13 to 12, an epimerization at C9 had occurred at some point during the lactonization step. Is the phraseology correct?

Line 237: "...in 74% yield.." Fig. 5 shows 78%.

Pg 13, Ln 262-268: The data in Table 2 can be summarized in the text and Table 2 can be moved to SI.

Pg 14, Ln 273-275: "... γ -hydroxylation of a mixture.....were unsuccessful..." this part is somewhat confusing: The first two reagents - IBX/DMSO, DBU/O₂ is γ -hydroxylation, but the latter two conditions correspond to γ -halogenation and sulfidation.

Line 284-285: "...THF or MeCN....led to recovery of the starting material." How much of the starting material was recovered?

Pg 17, Methods, Ln 349: "Yields refer to chromatographically." Sentence is incomplete.

Supplementary Information section

Pg S6: Synthesis of compound 15; equation shows 15 with R group but procedure describes the specific use of nBuMgBr. Following this on Pg S7, authors described a General Procedure for preparing 15a-d. The authors may want to reorganize this part, perhaps consolidating the procedure.

Pg S10: Compound 23a, but on page S68/69 it is numbered 23b.

Pg S18: "1H NMR and 13C NMR.....ung[3]." Capitalize Kende, Pyne and Ung.

REVIEWER COMMENTS

Reviewer #1 (Remarks to the Author):

In this manuscript, Huang and co-workers described their total syntheses of stemofoline and four related analogs including its plausible biogenetic precursors. This work is important and recommended for publication for the following reasons.

- i) Stemofoline and its analogs are important natural products with insecticidal and anticancer activity and they are very challenging target molecules for total synthesis.
- ii) Very few total syntheses of the closely related natural products were achieved. These prior total syntheses are elegant and remarkable, but there are also drawbacks in these syntheses. For example, the formation of the di-oxygenated tetrasubstituted double bond has been a long-standing challenge.
- iii) The synthetic approach described by Huang and co-workers are very different from the prior arts, therefore justifies for novelty.
- iv) They have developed a good solution to construct the problematic di-oxygenated tetrasubstituted double bond. Their protocol involves an HWE reaction followed by a bromine oxidation and a TBAF-promoted elimination and substitution. The HWE reaction allowed them to access two dihydrostemofoline analogs (**8** and **9**) and stemoburkilline (**7**), which could be further converted to stemofoline and isostemofoline. This result also provides evidence to support the proposed biogenetic hypothesis. This protocol will likely be adapted to make many other natural products with such a tetrasubstituted double bond in the stemona alkaloid family.
- v) A remarkable bridgehead bromide substitution via a sequence of elimination and addition is worth noting as well. This substitution allows the introduction of various alkyl group at the bridgehead carbon, which could be used to make other stemofoline analogs with varied alkyl group at this position.
- vi) Additionally, the authors identified a nice conjugate addition to solve the C10 stereochemistry problem, which troubled the previous syntheses.

With the above strength said, there are also a few minor weaknesses.

- i) Since the dihydrostemofoline analogs **8** and **9** and stemoburkilline (**7**) are inevitable intermediates on the way to stemofoline and isostemofoline is a minor product from the last step, the word “divergent” in the manuscript is not very accurate.

- ii) Is the stereochemistry of 23-1 and 23-2 different at C9 or C10? Why both can be converted to 24? An explanation is needed here.
- iii) Did the authors try any aerobic oxidation of 7, 8, or 9 to produce stemofoline?
- iv) The writing of this manuscript needs to be polished.

Reviewer #2 (Remarks to the Author):

The ms describes the asymmetric syntheses of four stemofoline-type alkaloids and a related diastereomer. The ms describes a substantial body of work aimed at synthesizing the tetracyclic core unit of the stemafoline ring system and then developing methodology to construct and install the butenolide onto the tetracyclic core. The linkage between the tetracyclic core and the butenolide moiety incorporates an ene diether motif.

Analytical and spectral data are consistent with structures of compounds.

Comments, questions, and clarifications which the authors might like to address are provided below.

Pg1, Abstract; ln 11-15: Sentence starts off in plural form and then changes to singular form.

Ln 17: "biogenic" or biogenetic hypothesis?

Ln 21; "...to tackle the frustrated ene diether connector"..... meaning of frustrated in this context??

Ln 26; ..."Stemona alkaloids having similar structural features." Useful to specify what the structural features are.....tetracyclic cage-type structure?

Pg 2, ln 52-55: "The attractive.....achieved." This section says almost the same thing as on pg 4, ln 83-84: "The intriguing....synthetic targets.", thus somewhat repetitive.

Pg 4, Fig. 1, structure 8: stereocentres at C-11 and C-12 can be better drawn. Since C-11 is part of a envelope conformation, the H and butenolyl unit is better represented as pseudoaxial and pseudoequatorial, respectively. The bond to the methyl group should be a normal line. Throughout the ms (and SI), bold bonds and normal bonds are used interchangeably to indicate stereochemistry.

Pg 5, ln 120-123: "...isostemofoline (2), stemoburkilline (7),.....DBU."This section is somewhat confusing as the authors talked about (11*S*,12*R*)-8 and mentioned Pyne's

work involving (11S,12S)-9 (Fig 2a). Further down, in 129-130, the same compounds were mentioned again. In Fig. 2A (pg 7), compound 9 is shown with (11R,12R) configurations. Further, the (11S,12S)-9 in Fig 5 added to the confusion.

Pg 7, Fig. 2A: The schematic showing the connectivity between compounds 7, 8, and 9 and alkaloids 1 and 2 is “oxidative cyclization”. Oxidative cyclization is applicable to 7  1 and 2; however, for 8/9 -- 1 and 2, the connection is “oxidation” (dehydrogenation).

Pg 8, Fig. 3: In the synthesis of tropinone 16, the % ratio of *cis*-17: *trans*-17 (60:21) was lower than the previous reported route (72:20; ref 55). Did the authors recycle *trans*-17 to obtain more *cis*-17 or was the *trans*-diastereomer not used?

Pg 9, Table 1(a) and (b): The way the data were presented caused some confusion at first, as the Table footnote b was easily mistaken by (b) (part of Table 1; in 201) depicting “Direct...other (functionalized) Grignard reagents” The presentation of structures 15a-d is somewhat awkward and the authors might want to combine their data into one Table.

The chemistry, modeled along the studies of Kibayashi et al., is very nice. Kibayashi's azabicyclic N,O-acetal substrates are of slightly larger ring sizes and the generation of anti-Bredt iminium ions are not as geometrically constrained. The authors' compound 16 is more structurally rigid and formation of the anti-Bredt iminium ion A (Table 1a) most likely is higher energy demanding. Do the authors think that the reaction might proceed via an organocopper-type intermediate – e.g., Tropinone-16—Cu(L)—Grignard instead of iminium ion A?

Pg 10-11 and Fig. 4: Line 206-207; The aldol reaction of tropinone 15 and glyoxalate 22 leads to aldol product 23-1 and 23-2 and a regioisomer 23a (not shown) The structure numbering used here is quite confusing as subsequent reactions indicating compound 23 to mean 23-1 and 23-2 (line 209). Also, 23a is labeled 23b in SI (S68/69). Does the 1.1:1 diastereomeric mixture indicate that the product is epimeric only at the carbinol center, and the α -carbon stereocentre is confirmed on account of the axial α -C-C(OH) bond?

Line 214: “...overall yield of 91.2%..” In Fig. 4, the yield shown is 95%.

Line 223: “The two transformations...83% yield.” Not exactly one step. The experimental procedure in SI suggests that the two steps were executed sequentially and involved a work-up step to obtain crude 27 which was used without purification in the lactonization step.

Line 225-226: "...which confirmed that....vicinal chiral centers at C9 and C10." In the conversion of 13 to 12, an epimerization at C9 had occurred at some point during the lactonization step. Is the phraseology correct?

Line 237: "...in 74% yield.." Fig. 5 shows 78%.

Pg 13, ln 262-268: The data in Table 2 can be summarized in the text and Table 2 can be moved to SI.

Pg 14, ln 273-275: "... γ -hydroxylation of a mixture.....were unsuccessful..." this part is somewhat confusing: The first two reagents – IBX/DMSO, DBU/O₂ is γ -hydroxylation, but the latter two conditions correspond to γ -halogenation and sulfidation.

Line 284-285: "...THF or MeCN.....led to recovery of the starting material." How much of the starting material was recovered?

Pg 17, Methods, ln 349: "Yields refer to chromatographically." Sentence is incomplete.

Supplementary Information section

Pg S6: Synthesis of compound 15; equation shows 15 with R group but procedure describes the specific use of ⁿBuMgBr. Following this on Pg S7, authors described a General Procedure for preparing 15a-d. The authors may want to reorganize this part, perhaps consolidating the procedure.

Pg S10: Compound 23a, but on page S68/69 it is numbered 23b.

Pg S18: "1H NMR and 13C NMR.....ung^[3]." Capitalize Kende, Pyne and Ung.

Reviewer #1

In this manuscript, Huang and co-workers described their total syntheses of stemofoline and four related analogs including its plausible biogenetic precursors. This work is important and recommended for publication for the following reasons.

- i) Stemofoline and its analogs are important natural products with insecticidal and anticancer activity and they are very challenging target molecules for total synthesis.
- ii) Very few total syntheses of the closely related natural products were achieved. These prior total syntheses are elegant and remarkable, but there are also drawbacks in these syntheses. For example, the formation of the di-oxygenated tetrasubstituted double bond has been a long-standing challenge.
- iii) The synthetic approach described by Huang and co-workers are very different from

the prior arts, therefore justifies for novelty.

iv) They have developed a good solution to construct the problematic di-oxygenated tetrasubstituted double bond. Their protocol involves an HWE reaction followed by a bromine oxidation and a TBAF-promoted elimination and substitution. The HWE reaction allowed them to access two dihydrostemofoline analogs (8 and 9) and stemoburkilline (7), which could be further converted to stemofoline and isostemofoline. This result also provides evidence to support the proposed biogenetic hypothesis. This protocol will likely be adapted to make many other natural products with such a tetrasubstituted double bond in the stemona alkaloid family.

v) A remarkable bridgehead bromide substitution via a sequence of elimination and addition is worth noting as well. This substitution allows the introduction of various alkyl group at the bridgehead carbon, which could be used to make other stemofoline analogs with varied alkyl group at this position.

vi) Additionally, the authors identified a nice conjugate addition to solve the C10 stereochemistry problem, which troubled the previous syntheses.

Our response: We are grateful for the visions, comments and questions, which are very helpful for us to improve this article.

With the above strength said, there are also a few minor weaknesses.

1. Since the dihydrostemofoline analogs 8 and 9 and stemoburkilline (7) are inevitable intermediates on the way to stemofoline and isostemofoline is a minor product from the last step, the word “divergent” in the manuscript is not very accurate.

Our response: Indeed, the original meaning of “divergent” synthesis involved the access to two or more products in equal amount. We intend to introduce this term to natural product chemistry/ photochemistry to account for the chemical diversity of natural product, those natural products are formed in non-equal amount. We believe that although living systems can produce each bio-molecules in an enzymatically controlled accurate manner (highly selective), the chemo-diversity of natural products might imply another strategy, namely, some steps of a synthesis deviate of enzymatic control. We shall elaborate this concept in a near future in form of a review/ perspective. Regarding isostemofoline, which is indeed a minor component among stemofoline alkaloid, has been the target of Kende’s total synthesis (ref. 48, J. Am. Chem. Soc. 1999, 7431). We are confident that the use of the word “divergent” is not accurate chemically, but bio/ phyto-chemically accurate.

2. Is the stereochemistry of 23-1 and 23-2 different at C9 or C10? Why both can be converted to 24? An explanation is needed here.

Our response: From the experimental results, we deduced that the structures of two diastereomers 23-1 and 23-2 have the stereochemistries shown the scheme. Subjecting of both diastereomers to derivatization with CDI and subsequent base-promoted anti-elimination then generated E-enone ester 24. The following sentences and structural details are added to the main text and Figure 4.

“From these experimental results, we deduced that the structures of two diastereomers 23-1 and 23-2 have the stereochemistries shown in Figure 4 (it became Fig. 5 for the revised version).”

3. Did the authors try any aerobic oxidation of 7, 8, or 9 to produce stemofoline?

Our response: For the oxidation of 7, we have tried several oxidation conditions including PdCl₂/CuCl/O₂ (cf. Pg 13 Ln 265 and the Supplementary Information, Table S1). For the oxidation of 8 and 9, several oxidation conditions have been tried including DBU/O₂ (cf. Pg 14 Ln 274 and the Supplementary Information, Table S2).

4. The writing of this manuscript needs to be polished.

Our response: Thanks for the kind suggestion. The manuscript has been sent for English polishing, but we didn't accept all corrections. The corrected manuscript has been further polished.

Reviewer #2

The ms describes the asymmetric syntheses of four stemofoline-type alkaloids and a

related diastereomer. The ms describes a substantial body of work aimed at synthesizing the tetracyclic core unit of the stemafoline ring system and then developing methodology to construct and install the butenolide onto the tetracyclic core. The linkage between the teracyclic core and the butenolide moiety incorporates an ene diether motif.

Analytical and spectral data are consistent with structures of compounds.

Our response: We are grateful for the visions, comments and questions, which are very helpful for us to improve this article.

Comments, questions, and clarifications which the authors might like to address are provided below.

1. Pg1, Abstract; Ln 11-15: Sentence starts off in plural form and then changes to singular form.

Our response: Thanks for indicating on this issue. The last sentence changes to plural form: “However, concise, enantioselective total syntheses of stemofoline alkaloids remain a formidable challenge due to their structural complexity.”

2. Ln 17: “biogenic” or biogenetic hypothesis?

Our response: Thanks for the question. In this text, “biogenic” changes to “biogenetic hypothesis”.

3. Ln 21; “...to tackle the frustrated ene diether connector”..... meaning of frustrated in this context??

Our response: “frustrated” changes to “challenging”.

4. Ln 26; ...”**Stemona** alkaloids having similar structural features.” Useful to specify what the structural features are.....tetracyclic cage-type structure?

Our response: To avoid any confusion, we deletes “having similar structural features”.

5. Pg 2, Ln 52-55: “The attractive.....achieved.” This section says almost the same thing as on pg 4, Ln 83-84: “The intriguing....synthetic targets.”, thus somewhat repetitive.

Our response: Thanks for indicating on this issue. On pg 4, Ln 83-84: “The intriguing....synthetic targets.” changes into “Much efforts have been devoted to the synthesis of stemofoline alkaloids.”

6. Pg 4, Fig. 1, structure **8**: stereocentres at C-11 and C-12 can be better drawn. Since C-11 is part of a envelope conformation, the H and butenolyl unit is better represented as pseudoaxial and pseudoequatorial, respectively. The bond to the methyl group should be a normal line. Throughout the ms (and SI), bold bonds and normal bonds are used interchangeably to indicate stereochemistry.

Our response: Thanks for the kind suggestions that also allowed us to correct the error on the stereochemistries at C11/C12. We have redrawn the chemical structures of related compounds according to your suggestions.

7. Pg 5, ln 120-123: “...isostemofoline (**2**), stemoburkilline (**7**),.....DBU.” This section is somewhat confusing as the authors talked about (11*S*,12*R*)-**8** and mentioned Pyne’s work involving (11*S*,12*S*)-**9** (Fig 2a). Further down, ln 129-130, the same compounds were mentioned again. In Fig. 2A (pg 7), compound **9** is shown with (11*R*,12*R*) configurations. Further, the (11*S*,12*S*)-**9** in Fig 5 added to the confusion.

Our response: We are in debt this reviewer for indicating this error. Indeed, in Fig. 2A, the absolute configuration of compound **9** [(11*R*,12*R*)] is wrong. It is corrected to **(11*S*,12*S*)** (cf. our response to item 6).

8. Pg 7, Fig. 2A: The schematic showing the connectivity between compounds **7**, **8**, and **9** and alkaloids **1** and **2** is “oxidative cyclization”. Oxidative cyclization is applicable to **7**  **1** and **2**; however, for **8/9** -- **1** and **2**, the connection is “oxidation” (dehydrogenation).

Our response: Thanks for indicating on this issue. In Fig. 2A “oxidative cyclization” changes to “oxidative cyclization or dehydrogenative oxidation”.

9. Pg 8, Fig. 3: In the synthesis of tropinone **16**, the % ratio of *cis*-**17**: *trans*-**17** (60 : 21) was lower than the previous reported route (72 : 20; ref 55). Did the authors recycle

trans-17 to obtain more *cis*-17 or was the *trans*-diastereomer not used?

Our response: Indeed, Yes, in the synthesis of tropinone 16, the *cis*-17: *trans*-17 ratio was lower than the previous route, but this approach to *cis*-17 is shortened by three steps as compared with previous one. According to our observation, *trans*-17 is the thermodynamically stable diastereomer, which could not be converted into *cis*-17, the *trans*-diastereomer not used.

10. Pg 9, Table 1 (a) and (b): The way the data were presented caused some confusion at first, as the Table footnote b was easily mistaken by (b) (part of Table 1; Ln 201) depicting “Direct...other (functionalized) Grignard reagents” The presentation of structures 15a-d is somewhat awkward and the authors might want to combine their data into one Table.

The chemistry, modeled along the studies of Kibayashi et al., is very nice. Kibayashi’s azabicyclic N,O-acetal substrates are of slightly larger ring sizes and the generation of anti-Bredt iminium ions are not as geometrically constrained. The authors’ compound 16 is more structurally rigid and formation of the anti-Bredt iminium ion A (Table 1a) most likely is higher energy demanding. Do the authors think that the reaction might proceed via an organocopper-type intermediate – e.g., Tropinone-16—Cu(L)—Grignard instead of iminium ion A?

Our response: Thanks for the kind suggestions. Now, Table 1a, b are separated into Table 1 and Fig. 4.

Thanks for helping analyzing the differences between Kibayashi’s ring system and ours. The generation of an organocopper-type intermediate is an interesting idea that is possible for the specific tropinone ring system as ours involving the formation of a homo-enolate. However, if it occurred, it should not be the main pathway because the reagents used herein for the reactions were Grignard reagents (nucleophiles) instead of electrophiles.

11. Pg 10-11 and Fig. 4: Line 206-207; The aldol reaction of tropinone 15 and glyoxalate 22 leads to aldol product 23-1 and 23-2 and a regioisomer 23a (not shown) The structure numbering used here is quite confusing as subsequent reactions indicating compound 23 to mean 23-1 and 23-2 (line 209). Also, 23a is labeled 23b in SI (S68/69). Does the 1.1:1 diastereomeric mixture indicate that the product is epimeric only at the carbinol center, and the α -carbon stereocentre is confirmed on account of the axial

α -C-C(OH) bond?

Our response: Thanks for indicating on these issues. First, “adduct 23” changes to “adducts 23-1 and 23-2”, and the structure of 23a provided. Second, in the Supplementary Information section, “23b” corrected to “23a”. Third, we were unable to distinguish between 23-1 and 23-2 by their NMR spectra. From the experimental results, we deduced that the structures of two diastereomers 23-1 and 23-2 have the stereochemistries shown the scheme. Subjecting of both diastereomers to derivatization with CDI and subsequent H base-promoted anti-elimination then generated E-enone ester 24. The following sentences and structural details are added to the main text and Figure 4.

“From these experimental results, we deduced that the structures of two diastereomers 23-1 and 23-2 have the stereochemistries shown in Figure 4 (it became Fig. 5 for the revised version).”

12. Line 214: “...overall yield of 91.2%.” In Fig. 4, the yield shown is 95%.

Our response: 91.2% is the overall yield for two steps. “91.2%” changes to “91.2% for two steps”. Thanks for indicating the problem.

13. Line 223: “The two transformations....83% yield.” Not exactly one step. The experimental procedure in SI suggests that the two steps were executed sequentially and involved a work-up step to obtain crude 27 which was used without purification in the lactonization step.

Our response: This involved one operation and two pots. We change it to two steps. In the text, “in a single step” (pg11 ln223) and “in a one-step manner” (pg 12 ln 233) change to “Importantly, neither the hydrogenation or lactonization step required purification to yield 12 of sufficient purity for further reactions.” And the total steps

changes accordingly throughout the manuscript from 18 steps to 19 steps.

14. Line 225-226: “...which confirmed that...vicinal chiral centers at C9 and C10.” In the conversion of 13 to 12, an epimerization at C9 had occurred at some point during the lactonization step. Is the phraseology correct?

Our response: We have redrawn the structures of compounds 13 and 27 according to the reviewer's suggestions (see below). In addition, to confirm the relative stereochemistry of compound 13, its NOE spectrum was recorded (and the relative stereochemistry confirmed) and included in SI (Supplementary Figure S6 and S7, pg S40-41). Compound 13 and compound 12 have the same chiral center at C9.

15. Line 237: “...in 74% yield..” Fig. 5 shows 78%.

Our response: Thanks for indicating our error. “74%” changes to “78%”.

16. Pg 13, ln 262-268: The data in Table 2 can be summarized in the text and Table 2 can be moved to SI.

Our response: Thanks for the kind suggestions. Table 2 removed to Supplementary Information (Supplementary Table S1, pg S23).

17. Pg 14, ln 273-275: “... γ -hydroxylation of a mixture.....were unsuccessful...” this part is somewhat confusing: The first two reagents – IBX/DMSO, DBU/O₂ is γ -hydroxylation, but the latter two conditions correspond to γ -halogenation and sulfidation.

Our response: Thanks for indicating on this issue, “ γ -hydroxylation” changes to “oxygenation with IBX/DMSO or DBU/O₂, halogenation with NIS/NBS, or methylsulfidation with MeSSMe”.

18. Line 284-285: “...THF or MeCN....led to recovery of the starting material.” How much of the starting material was recovered?

Our response: More than 95% of the starting material was recovered. Now, we add

“However, 7 apparently showed no reaction with NIS and was thus mostly recovered (>95%)...” in the text (pg14 ln285).

19. Pg 17, Methods, ln 349:” Yields refer to chromatographically.’ Sentence is incomplete.

Our response: “Yields refer to chromatographically.” changes to “Yields were calculated based on the weights of chromatographically isolated products.”

20. Supplementary Information section

Pg S6: Synthesis of compound 15; equation shows 15 with R group but procedure describes the specific use of ⁿBuMgBr. Following this on Pg S7, authors described a General Procedure for preparing 15a-d. The authors may want to reorganize this part, perhaps consolidating the procedure.

Our response: Thanks for indicating on this issue. In the equation regarding the synthesis of compound 15, “R” changes to “ⁿBu”.

21. Supplementary Information section

Pg S10: Compound 23a, but on page S68/69 it is numbered 23b.

Our response: Thanks for indicating the error, “23b” is corrected to “23a”.

22. Supplementary Information section, Pg S18: “¹H NMR and ¹³C NMR.....ung^[3].” Capitalize Kende, Pyne and Ung.

Our response: Corrected with thanks.

REVIEWERS' COMMENTS:

Reviewer #1 (Remarks to the Author):

This authors have addressed this reviewer's comments. The manuscript is recommended for acceptance.

Reviewer #2 (Remarks to the Author):

The revisions to the ms is satisfactory and has clarified/addressed issues and questions that have been raised. Publication is recommended.

However, I still have my concerns regarding the use of the word "divergent". I appreciate the intention of the authors to provide support to their biogenetic hypothesis. Nonetheless, the ms describes the total syntheses of the said alkaloids and the production of the desired alkaloids was realized through reactions that were chemically non-selective. I do not think that the non-selective reactions were by design or intentionally planned by the authors.

Leaving "divergent" in the title and elsewhere would lead readers to think that the reported work is describing bona fide divergent syntheses of the alkaloids.

Point-by-point response to referees

REVIEWERS' COMMENTS:

Reviewer #1 (Remarks to the Author):

This authors have addressed this reviewer's comments. The manuscript is recommended for acceptance.

Our response: We are grateful for the comments.

Reviewer #2 (Remarks to the Author):

The revisions to the ms is satisfactory and has clarified/addressed issues and questions that have been raised. Publication is recommended.

However, I still have my concerns regarding the use of the word “divergent”. I appreciate the intention of the authors to provide support to their biogenetic hypothesis. Nonetheless, the ms describes the total syntheses of the said alkaloids and the production of the desired alkaloids was realized through reactions that were chemically non-selective. I do not think that the non-selective reactions were by design or intentionally planned by the authors.

Leaving “divergent” in the title and elsewhere would lead readers to think that the reported work is describing bona fide divergent syntheses of the alkaloids.

Our response: Thanks for the kind suggestions, we have deleted all “divergent”.